# The Factorization Curse: Which Tokens You Predict Underlie the Reversal Curse and More

**Ouail Kitouni**[*1]**, Niklas Nolte** [2]**, Diane Bouchacourt** [2]**, Adina Williams** [2]**, Mike Rabbat** [2]**, and Mark Ibrahim** [2]

[1]IAIFI, Massachusetts Institute of Technology
[1]FAIR at Meta

## Abstract

Today's best language models still struggle with *hallucinations*: factually incorrect generations, which impede their ability to reliably retrieve information seen during training. The *reversal curse*, where models cannot recall information when probed in a different order than was encountered during training, exemplifies this in information retrieval. We reframe the reversal curse as a *factorization curse* — a failure of models to learn the same joint distribution under different factorizations. Through a series of controlled experiments with increasing levels of realism including *WikiReversal*, a setting we introduce to closely simulate a knowledge intensive finetuning task, we find that the factorization curse is an inherent failure of the next-token prediction objective used in popular large language models. Moreover, we demonstrate reliable information retrieval cannot be solved with scale, reversed tokens, or even naive bidirectional-attention training. Consequently, various approaches to finetuning on specialized data would necessarily provide mixed results on downstream tasks, unless the model has already seen the right sequence of tokens. Across five tasks of varying levels of complexity, our results uncover a promising path forward: factorization-agnostic objectives can significantly mitigate the reversal curse and hint at improved knowledge storage and planning capabilities.

## 1 Introduction

Although today's best language models produce impressively cogent, articulate text by learning the statistics of language, they still struggle to reliably retrieve information seen during training. Models are known to suffer from hallucinations, potentially responding with fabricated content that differs from the knowledge present in training data. Hallucinations pose a significant hurdle to the adoption of language models, especially in domains where reliable knowledge retrieval is paramount (Dahl et al., 2024). A well-studied failure mode underlying hallucinations is the *reversal curse*, which ascribes this deficiency to the precise order of words presented to the model at train-time (Berglund et al., 2023; Allen-Zhu & Li, 2023). For example, a model trained on sentences where *Paris* always appears as the subject of the sentence, such as *"Paris is the capital of France"*, can be tuned to answer *"Paris is the capital of which country?"* but not *"What is the capital of France?"*, even though these two formulations encode the same underlying information. Existing approaches aimed at mitigating the reversal curse have focused on data augmentations that involve training on both the forward and reversed tokens (Golovneva et al., 2024). In this work, we focus on learning objectives.

In Section 2, we propose the *factorization curse*, a framework that characterizes the reversal curse as a failure to model the same joint distribution under different factorizations. We show the prevailing

---

[*]Work done while interning at FAIR, Meta.

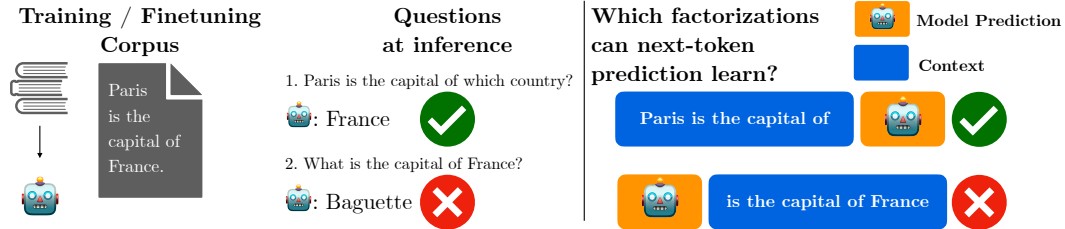

Figure 1: **(Left)** Reversal curse from training a model on sentences with *Paris* before *France*. **(Right)** Left-to-right objective does not learn how to predict early tokens from later ones even if the information content is the same. The model overfits to a specific factorization of the joint distribution over tokens, and is unable to answer questions that require reasoning about a different factorization.

left-to-right next token prediction, autoregressive (AR) objective used in popular large models such as GPT (Radford et al., 2019) and Llama models (Touvron et al., 2023a,b), underlies the reversal curse. We illustrate in Figure 1 how the factorization in AR training only encodes information based on prior context, thereby limiting how well the model can retrieve information based on *later context*. Through this lens, we show the reversal curse is not merely a failure to learn logical implications, but a more general problem related to learning objectives. Given this framework, we hypothesize in Section 2.1 that factorization agnostic models, *i.e.*, models trained in a manner that is less dependent on the specific token order while preserving the overall meaning, can store knowledge better and are less prone to the reversal curse. To validate our hypothesis and explore potential solutions, we conduct extensive experiments in controlled settings in Section 3.1, focusing on the effects of pretraining objectives on knowledge storage. Section 3.2 introduces *WikiReversal*, a realistic testbed based on Wikipedia knowledge graphs that closely replicates a knowledge-intensive finetuning application. In experiments with increasing levels of complexity and realism, we observe that scale, naive bidirectional objectives, and even left-to-right training do not resolve the reversal curse. These results suggest that finetuning strategies for downstream applications might not allow models to store knowledge adequately. Finally, in Section 4, we find that factorization-agnostic training is not only a promising initial solution for knowledge storage but also hints at improved planning capabilities in a minimal graph traversal task.

To summarize, our contributions are as follows:

1. We introduce the concept of the *factorization curse*, which posits that different objectives' decomposition of an input into context and prediction is a key factor underlying the reversal curse.

2. We conduct empirical studies with increasing levels of complexity and realism to validate our framework, comparing strategies such as standard autoregressive training (AR), AR with reversed sequences, and masked language modeling (MLM) as a prototypical bidirectional objective.

3. Building on our factorization curse framework, we identify factorization-agnostic objectives that allow for making predictions using every possible context decomposition, as a strong baseline solution. We explore its effectiveness across all investigated settings, including the WikiReversal setting.

4. We show that factorization-agnostic strategies are promising not only for knowledge storage/retrieval but also for planning, suggesting potentially broader implications for our findings.

## 2 The Factorization Curse

The reversal curse highlights how language models struggle to reliably retrieve information seen during training given some context. Our aim is to understand this failure by probing how common training objectives factorize an input into context and prediction. We show how common training objectives, including popular left-to-right AR and masked modeling objectives, struggle to learn factorizations that can help the model generalize better on a given task, a challenge we label the *factorization curse*.

## 2.1 Hypothesis: Reversal Curse as an Instance of Factorization Curse

Let us define the *factorization curse* more formally. We first start with the usual left-to-right autoregressive model for a sequence $\boldsymbol{x}$ with $D$ tokens. This is the standard formulation in popular GPT-style (Radford et al., 2019; OpenAI, 2023) models and its loglikelihood is given by

$$\log p(\boldsymbol{x}) = \sum_{t=1}^{D} \log p(x_t|\boldsymbol{x}_{<t}). \tag{1}$$

Each token is represented as $x_t$, where $t$ is its index in the sequence. $\boldsymbol{x}_{<t}$ represents all tokens that precede the $t$-th token in the sequence. The log probability of the entire sequence $\boldsymbol{x}$ is computed as the sum of the log probabilities of each token $x_t$, given all the preceding tokens $\boldsymbol{x}_{<t}$. This is the *left-to-right factorization* of the joint distribution over tokens. Note that there are many factorizations ($D!$) of the joint distribution, each given by some permutation $\sigma$ of the tokens in the sequence, which we can write as $\log p(\boldsymbol{x}) = \sum_{t=1}^{D} \log p(x_{\sigma(t)}|\boldsymbol{x}_{\sigma(<t)})$.

**Example in Two Tokens** For illustration purposes, let us walk through an example with $D = 2$. Suppose our goal is to model $p(\boldsymbol{x}) = p(x_2, x_1) = p(x_2|x_1)p(x_1)$. The left-to-right factorization loss optimizes

$$-\mathcal{L}_{AR} = \log p(\boldsymbol{x}) = \log p(x_2|x_1) + \log p(x_1). \tag{2}$$

Interestingly, we can readily see the reversal curse failure mode in $\mathcal{L}_{AR}$. A model $p_\theta$ that attributes high likelihood to $p_\theta(x_2|x_1)p_\theta(x_1)$ does not necessarily yield a high value for $p_\theta(x_1|x_2)p_\theta(x_2)$ (a right-to-left factorization) even though the two expressions should be equal due to the chain rule of probability. Note that here we make no statement about the sequential order of the random variables or their relationship. The only statement we make is that, unsurprisingly, $p_\theta$ is not necessarily capable of modeling the joint distribution when presented with a different factorization. This is the *factorization curse*.

**Definition 1** *(Factorization Curse). A model $p_\theta$ for the joint distribution of a sequence $\boldsymbol{x}$ suffers from the factorization curse if, for a factorization order $\sigma$ different from the "training" factorization order $\sigma_0$ (which depends on the objective and model details), we have*

$$\prod_t p_\theta(x_{\sigma(t)}|x_{\sigma(<t)}) \neq \prod_t p_\theta(x_{\sigma_0(t)}|x_{\sigma_0(<t)}). \tag{3}$$

*In particular, the model may be optimal on $\sigma_0$, but perform poorly on a different factorization.*

**Implications** This has a number of implications. First, by definition, a highly factorization-dependent LLM will struggle to retrieve knowledge from earlier in the context given later information. Second, simply training on additional sequences with all tokens reversed would likely not alleviate the issue. Indeed, if the information we seek to retrieve is composed of multiple tokens, the factorization the LLM needs to handle is not right-to-left, but instead reversed in chunks. Thus, in order for any reverse training strategy to work, one must first parse the entities of interest then train with sequences reversed in entity-preserving chunks (see Section 3 and Figure 6).

Furthermore this explains why standard MLM approaches with fixed masking rates fail to address the issue, despite their bidirectionality, for two reasons: First, entities may span a larger number of tokens than the model masks, meaning there is never supervision signal to make the prediction from the right context (without leaking parts of the entity). Second, training with a fixed rate does not yield a good generative model. While the model is used to predicting, *e.g.*, 15% of a full context-window during training, at inference, the model can fail to generalize (Tay et al., 2022) to the arbitrary-length sequences it encounters (see Figure 2). Zhang et al. (2024) suggest that encountering different length sequences during training encourages disentanglement and compositionality, which will be crucial for a good generative model.

**Knowledge retrieval beyond reversal:** A model that cannot learn how to retrieve information in reverse order will likely suffer from further downstream issues that are often ascribed to hallucination. For instance, let us take a model pretrained on entities in a database, say a list of soccer players with various attributes (statistics, game histories, *etc.*) with the name appearing before the attributes as follows $\boldsymbol{x}_{\text{name}}, \boldsymbol{x}_{\text{attributes}}$. The model may memorize the database perfectly, but when queried

for players that match specific criteria (*e.g.*, played in a particular position, or have a specific nationality, *etc.*), the model can produce hallucinated answers that do not match the training distribution due to lack of direct supervision of the form $p(\boldsymbol{x}_{\text{name}}|\boldsymbol{x}_{\text{attributes}})$ during pretraining.

## 2.2 Factorization-Agnostic Training Strategies

To store and retrieve knowledge in "all directions" for arbitrary-length entities and without external intervention (entity pre-parsing, retrieval augmented generation, *etc.*), the model needs to be equally good at any factorization of the joint distribution. Below, we discuss two ways this can be achieved.

**Permutation Language Modeling (PLM)** A straightforward way to alleviate the factorization issue, is to write the autoregressive loss in a way that is independent of factorization by averaging over all permutations as follows

$$\log p(\boldsymbol{x}) = \log \mathbb{E}_{\sigma \sim \mathcal{U}(S_D)} \left[ \prod_{t=1}^{D} p(x_{\sigma(t)}|\boldsymbol{x}_{\sigma(<t)}) \right] \geq \mathbb{E}_{\sigma \sim \mathcal{U}(S_D)} \left[ \sum_{t=1}^{D} \log p(x_{\sigma(t)}|\boldsymbol{x}_{\sigma(<t)}) \right], \quad (4)$$

where $\sigma$ is a permutation sampled uniformly at random from $S_D$, the permutation group on $D$ tokens. The term $\boldsymbol{x}_{\sigma(<t)}$ represents all tokens that precede the $t$-th token in the permuted sequence. The log probability of the entire sequence $\boldsymbol{x}$ is then lower-bounded (using Jensen's inequality) by the expected sum of the log probabilities of each element $x_{\sigma(t)}$, given all its preceding tokens in the permuted sequence. Note that all we did here is average over all factorizations. This formulation is used in XLNet (Yang et al., 2020). However, for practical reasons they end up training with a permutation on the last few tokens only. This partial prediction, as we argued above, can limit knowledge storage improvements because we do not know how to chunk tokens into entities a priori.

**Uniform-Rate Masked Language Modeling (MLM-$\mathcal{U}$)** An alternative factorization-agnostic objective is to predict any context from any other context uniformly at random. This includes next-token, previous-token, predictions spanning multiple future or past tokens, and all other forms of contextual prediction. As it turns out, this generalization over objectives (amounting to something similar to masked language modeling with a randomly sampled masking rate $r \sim \mathcal{U}(0,1)$) is a discrete diffusion model with an absorbing masking state (Austin et al., 2023; Kitouni et al., 2024). This diffusion formulation can be used to make a factorization-order-independent autoregressive model. See Figure 2 for an illustration of the differences between the MLM-$\mathcal{U}$ objective and more standard MLM. Specifically, $\mathcal{L}_{CT}$ from Kitouni et al. (2024)'s Proposition 1, which we will refer to as $\mathcal{L}_{\text{MLM-}\mathcal{U}}$ here, can be retrieved from Equation (4) as follows

$$
\begin{aligned}
\mathcal{L}_{\text{MLM-}\mathcal{U}} &= -\mathbb{E}_{\sigma \sim \mathcal{U}(S_D)} \sum_{t=1}^{D} \log p(x_{\sigma(t)}|\boldsymbol{x}_{\sigma(<t)}) \\
&= -\mathbb{E}_{\sigma \sim \mathcal{U}(S_D)} \mathbb{E}_{t \sim \mathcal{U}(1,\cdots,D)} D \log p(x_{\sigma(t)}|\boldsymbol{x}_{\sigma(<t)}) \\
&= -\mathbb{E}_{\sigma \sim \mathcal{U}(S_D)} \mathbb{E}_{t \sim \mathcal{U}(1,\cdots,D)} \frac{D}{D-t+1} \sum_{\tau \in \sigma(\geq t)} \log p(x_{\tau}|\boldsymbol{x}_{\sigma(<t)}) \quad (5)
\end{aligned}
$$

where the last equality is possible because all $\tau \in \sigma(\geq t)$ tokens are equally likely to appear at position $t$ as we average across all permutations, and so we can average over the predictions for each $\tau$ at this position. This approach can be implemented as a denoising process which recovers randomly masked tokens, like BERT (Devlin et al., 2019), but with uniformly sampled masking rates. This key difference allows training a generative model with masked modeling.[2]

## 3 Experiments

We now investigate information retrieval capabilities across learning objectives through the lens of different factorizations of the joint sequence probability. Specifically, we compare

---

[2]Appendix B shows a simple example illustrating the connection to permutation language modeling.

- **AR:** The standard autoregressive causal next-token prediction. Though all models generate tokens autoregressively, we use AR as a shorthand for left-to-right models, in line with Equation (1).
- **AR w/reverse:** AR prediction on sequences and their token-level reverse.[3]
- **MLM $r$:** BERT-like masked language modeling with fixed random masking rate, $r$.
- **MLM-$\mathcal{U}$:** MLM with $r \sim \mathcal{U}(0, 1)$. PLM results are similar, and are reported in the Appendix.

To ensure a fair comparison and allow each objective to perform optimally, we employ model architectures specifically designed for each objective. For autoregressive (AR) training, we use GPT-2 (Radford et al., 2019) and Mistral (Jiang et al., 2023). For masked language modeling (MLM), we use BERT (Devlin et al., 2019). Finally, for MLM-$\mathcal{U}$, we employ an encoder-decoder model[4] based on the GPT architecture (see Appendix G for details).

We study these models across increasing levels of complexity and realism, beginning with a controlled retrieval task using synthetic tokens to a retrieval task using natural text from Wikipedia articles. In our evaluation, we find that the degree to which the learning objective factorizes the input reliably explains performance across this wide range of information retrieval tasks. Factorization-agnostic methods show improved knowledge retrieval capabilities.

### 3.1 Controlled Experiments in Factorization-Agnostic Training

**Retrieval Task.** We are particularly interested in models' ability to recall knowledge from data they were trained on. We will use a simple toy task, adapted from Golovneva et al. (2024), to evaluate this capability. First, we generate a collection of key-value pairs which are each composed of a sequence $\{t_i\}^{i \in S}$ of tokens, *e.g.*, consider the key-value pair

$$t_0 t_1 : t_2 t_3.$$

Each key/value is unique and is composed of a unique set of tokens to control for any effects due to token statistics. Additionally, we generate two types of queries: (forward) "[value of] *key* : *value*" and (backward) "[key of] *value* : *key*". Models are trained on all key-value pairs and a subset of queries, and tested on unseen queries by completing tokens after the colon. Accuracy, measured using exact match, in Table 1 shows AR training does not retrieve keys from values and that reversing tokens does not improve backward retrieval. We observe entity-based reversing trivially solves this task. Additionally, while MLM does not suffer from a forward/backward disparity, its fixed masking rate causes poor overall results. Introducing a uniformly sampled rate via MLM-$\mathcal{U}$ solves the task perfectly.

**Non-reciprocal Relationships. Are models employing incorrect reversal heuristics?** A weakness of the retrieval task is that it could be solved by assuming all relations between keys and values to be symmetric/reciprocal. In language, this is not always the case: even though they contain many of the same words, the sentence *Alice likes Bob* does not necessarily imply that *Bob likes Alice*. To investigate whether models inappropriately rely on a reversal heuristic, we extend the retrieval task

---

[3]We obtained similar results when manipulating attention masks to train on an equal mix of causal and "anti-causal" sequences.

[4]We also ran our experiments with this architecture for all the objectives and found consistent results.

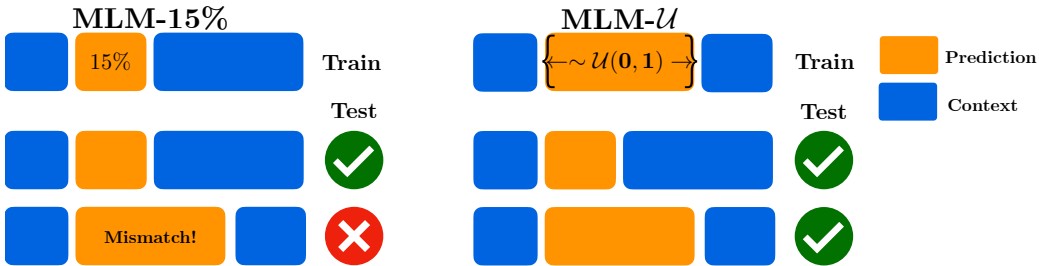

Figure 2: MLM struggles when entities span more tokens than the masked span. MLM-$\mathcal{U}$ encounters all possible masking fractions during training and does not suffer from this problem.

Table 1: Exact match accuracy of different training paradigms on **(Top)** the retrieval task and **(Bottom)** relationship task. Due to the non-reciprocal nature of the relationship, a model that swaps the subject and object will make errors (e.g., inferring $B$ is $A$'s child from $A$ being $B$'s child). Shown in the bottom row. Entity reversal without a delimiter is marked with a*. Maximum values are bold.

| Retrieval Task | AR | AR w/reverse | MLM | MLM-$\mathcal{U}$ |
|---|---|---|---|---|
| Forward ↑ | **100** | **100** | 21 | **100** |
| Backward ↑ | 0 | 0 | 22 | **100** |

| Relationship Task | AR w/reverse (entity)* | AR w/reverse (entity) | MLM | MLM-$\mathcal{U}$ |
|---|---|---|---|---|
| Forward ↑ | 54 | **100** | 24 | **100** |
| Backward ↑ | 53 | **100** | 19 | **100** |
| Incorrect Inference ↓ | **44** | 0 | 0 | 0 |

to a third entity for each sample, yielding statements of the form "$A \implies B \implies C$". Question answering (QA) samples are of the form (forward) "$B \implies ?$" and (backward) "$B \impliedby ?$" where the right answers are $C$ and $A$, respectively. With a third entity in play, a model assuming symmetry would be unable to decide between $A$ and $C$ as the answer for either question.

The bottom of Table 1 shows that simply reversing entities (denoted with AR w/reverse (entity)*) leads to undesirable behaviour as can be seen from the large incorrect inference rate. However, adding simple delimiter tokens around entity reversed sequences (without asterisk) leads to more a robust model. Finally, MLM-$\mathcal{U}$ learns the asymmetric relations correctly.

**BioS.** Next, we investigate performance of the different objectives for more complex but still controlled data. BioS (Zhu & Li, 2023) is a synthetic dataset consisting of biographies for 10k fictional individuals. For each individual, biographical details (birth date, birth city, *etc.*) were randomly selected from a uniform distribution. The authors ensured each individual was assigned a unique full name. We reproduce some of their results on the `birth_date-to-full_name` task which aims to recover a person's full name from their birth date. Results are shown in Table 2. Again, the autoregressive, MLM and reversed-token training struggle to recover backward queries.

Training in a factorization-agnostic fashion leads to non-negligible backward performance. Interestingly, backward performance keeps improving over a long time (many times the number of epochs required for forward performance to reach 100%) (see Appendix F). If this delay is due to the low frequency of observing the right factorization in training, this could indicate that methods to automatically select data such as RHO-LOSS (Mindermann et al., 2022) could have a disproportionate impact in improving factorization-agnostic methods compared to standard AR training.

Table 2: BioS exact match accuracy for property retrieval in the backward direction (birth date to full name) and in the forward direction (full name to birthdate).

| | AR | AR w/reverse | MLM | MLM-$\mathcal{U}$ |
|---|---|---|---|---|
| Forward | 100 | 100 | 8 | 100 |
| Backward | 0 | 0 | 0 | 68 |

### 3.2 Wikipedia Knowledge Graph Reversal

To bridge the gap between the controlled studies on synthetic datasets and more realistic evaluations, we introduce a new evaluation setup that combines the best of both approaches. Our setup involves finetuning a language model on real-world natural text from Wikipedia articles, along with a precise knowledge graph describing the relations and entities within them. This allows for principled experiments that mimic real-world use-cases where practitioners finetune pretrained models on domain-specific corpora. We compare finetuning a language model in this standard setup to training from scratch with MLM-$\mathcal{U}$.

**Experiment Design** We introduce a new closed-book QA dataset to evaluate the ability of models to reason about entities and relations in both forward and backward directions. The dataset is derived

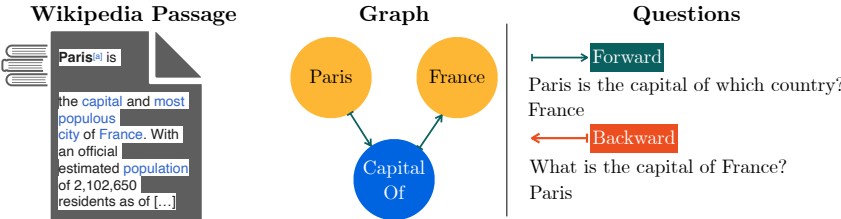

Figure 3: An example passage with a forward relation triple. The forward question queries the tail, backward queries the head. *WikiReversal* is a collection of passages and forward/backward QAs.

from the GenWiki corpus based on DBpedia (Jin et al., 2020), which contains 1.3 million text passages from Wikipedia, along with entity and relation annotations.

**Extracting Relation Triples and Generating Questions**   For each passage $P$ with annotated entities $E = e_1, e_2, ..., e_n$, we consider only "forward" relation triples $(e_i, r, e_j)$, where $e_i$ appears before $e_j$ in the passage. For the example, in the passage *"Paris is the [...] capital of France [...]"* (Figure 3), the triplet (*Paris*, *capitalOf*, *France*) is a "forward" triplet. Had the triplet (*France*, *hasCapital*, *Paris*) been present in the graph, we would consider it a "backward" triplet. We filter the data to contain only triplets (and corresponding passages) for which the relation $r$ exists at least in 500 different instances. We generate forward questions $F_r(e_i)$ querying the tail of the relation $(e_j)$ and backward questions $B_r(e_j)$ querying the head $(e_i)$ using predefined templates. We filter out ambiguous samples to ensure each question has a single unique answer. Algorithm 1 in the Appendix summarizes the dataset creation process.

Table 3 reports the forward/backward performance disparity, particularly for autoregressive models. Mistral 7B, achieves a backward accuracy of around 5%, much lower than its forward accuracy. Interestingly, the model starts around the same backward accuracy in the beginnig of finetuning. This indicates there may still be backwards triplets present in a "forward fashion" within the model's training text. This could also explain the non-trivial back-

Table 3: Wikireversal task exact match QA accuracies. MLM-$\mathcal{U}$, MLM and AR are are 100M parameter models trained from scratch.

|  | Mistral 7B | MLM | MLM-$\mathcal{U}$ | AR |
|---|---|---|---|---|
| Forward | **21** | 3.4 | 11 | 14 |
| Backward | 5.2 | 2.7 | **7.9** | 4.3 |

ward performance of the AR model, despite its susceptibility to the reversal curse. MLM-$\mathcal{U}$ attains the highest backward accuracy among the evaluated models, demonstrating its robustness to the reversal curse. This suggests training from scratch with MLM-$\mathcal{U}$ can outperform a much larger ($70 \times$) pretrained language model finetuned on the same data. However, it still falls short of the AR model's forward performance, possibly due to the inherent difficulty of the task. Notably, significantly better results can be obtained by allowing models to leverage knowledge stored from the QAs themselves (see Appendix E.3 for details).

## 3.3   Analyzing Representations Learned via Factorization-Agnostic Training

We further examine factorization-agnostic training by first comparing the role of random masking in MLM-$\mathcal{U}$ versus standard masked language modeling. We also visualize the learned representations from MLM-$\mathcal{U}$ showing they contain more distinct entity structure compared to standard AR training.

**Understanding the role of random masking**   To understand the importance of varying the masking ratio as introduced in MLM-$\mathcal{U}$ we compare MLM-$\mathcal{U}$ to MLM with various masking ratios (15%, 40%, 85%) based on prior work (Wettig et al., 2023)). We find MLM exhibits much noisier performance that's consistently lower than MLM-$\mathcal{U}$ with uniformly random masking ratio as shown in Figure 4a. This suggests fixed masking ratios, whether with high or low values, are limited in what they can learn in contrast to MLM-$\mathcal{U}$.

**Visualizing learned representations in the 3-entity relationship task**    To better probe the representations learned via MLM-$\mathcal{U}$ we plot in Figures 4b and 4c the PCA projections after training on the relationship task from Section 3.1 for AR and MLM-$\mathcal{U}$. Compared to AR, which learns disconnected components without apparent symmetry for entities never seen backwards during training, MLM-$\mathcal{U}$ seems to have learned a form of translation symmetry across train and test samples. This suggests MLM-$\mathcal{U}$ training leads to more structured entities in the model's representation space.

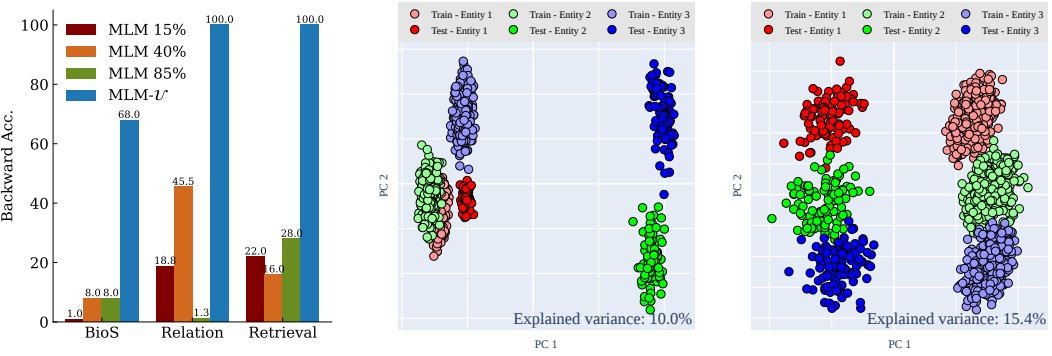

(a) Fixed-rate masked modeling is inconsistent.

(b) AR model entities PCA projection.

(c) MLM-$\mathcal{U}$ model entities PCA projection.

Figure 4: In panel (a) we compare MLM with varying masking ratios to MLM-$\mathcal{U}$. In panels (b) and (c) we visualize the two main principal components of representations learned via AR versus MLM-$\mathcal{U}$.

We also measure the computational efficiency relative to convergence of MLM-$\mathcal{U}$ versus autoregressive (next token) training showing favorable efficiency for MLM-$\mathcal{U}$ in Appendix Figure 9.

# 4    On the Importance of Future Predictions for Planning

Prior work argues next-token prediction auto-regressive loss is not conducive to planning (Dziri et al., 2023; LeCun, 2023; Gloeckle et al., 2024). Specifically, Bachmann & Nagarajan (2024) introduces a simple path finding task that requires basic planning: From a start node, multiple linear paths $p_1, p_2, \ldots, p_n$ extend outward. They are given as symbolic sequences of this form: $\underbrace{2,6|6,7|5,1|4,3|4,2|3,5}_{\text{edges}}$ / $\underbrace{4,7}_{\text{start,end}}$ = $\underbrace{4,2,6,7}_{\text{desired response}}$    A model is tasked to predict the sequence of nodes along path $p_i$ that leads to a specified final node at the end of $p_i$. They show that when trained with a standard autoregressive (AR) next-token prediction objective, the model is unable to effectively learn this task. This failure is attributed, at least in part, to the teacher-forcing supervision used during training. As illustrated in Figure 5, from the second node $x_2 = 2$ onward along a path $p_i = (x_1, x_2, \ldots, x_m)$, the model can predict each "easy" token $x_t$ for $t > 2$ by simply conditioning on the immediately previous teacher-forced token $x_{t-1}$, without requiring retention of the earlier path history or look-ahead planning, a pitfall referred to as the "Clever Hans" cheat (see Section 4.5 (Bachmann & Nagarajan, 2024) and (Pfungst & Rosenthal, 1911)).

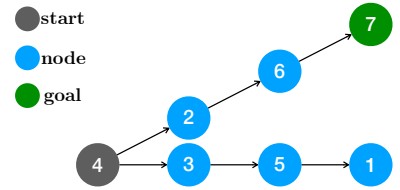

(a) Accuracies of various training paradigms on the Star Graph Task. Randomly choosing a starting node in this setting (and employing the Clever Hans Cheat) results in 50% accuracy.

|  | AR | AR w/reverse | MLM-$\mathcal{U}$ |
|---|---|---|---|
| Accuracy | 50 | 49 | **100** |

Figure 5: Star Graph Task: Illustration and Performance Comparison. The illustration shows the "Clever Hans" failure mode with teacher-forced AR ((Bachmann & Nagarajan, 2024) adapted).

Bachmann & Nagarajan (2024) found that predicting multiple future tokens in a teacher-less setting helped mitigate the issue of discovering the algorithm to correctly predict the initial "difficult" token $x_2$. We identify this as an intermediate objective between standard next-token prediction and the factorization-agnostic objective studied in this work, which encourages planning capabilities via both far look-ahead and look-backward along the sequence. Figure 5a shows that the MLM-$\mathcal{U}$ objective enables the model to reliably solve the path-planning task by better capturing the planning requirements.

## 5 Related Work

The reversal curse was first introduced in Berglund et al. (2023). Using text-overlap based heuristics for modeling inferences between sequence of text dates back nearly two decades in NlP (Glickman et al., 2005; Adams et al., 2007). As our modeling approaches have improved, increasing work has drawn attention to models overapplying text-overlap heuristics (Dasgupta et al. 2018; Naik et al. 2018; Sanchez et al. 2018; McCoy et al. 2019; Rajaee et al. 2022; Williams et al. 2022, i.a.). Perhaps most relevant is Sinha et al. (2019)'s evaluation, which used synthetic entity-based kinship data with multiple entities based on graph structures to expose model failures and is similar to our relationship task. Most recently, work aimed at mitigating the reversal curse by Allen-Zhu & Li (2023); Golovneva et al. (2024) suggest using data augmentations by reversing both token sequences, or if available, entity orders by training both on the forward and augmented text. Related projects have also trained and/or finetuned RoBERTa (Liu et al., 2019) or BERT (Devlin et al., 2019)-based models on input sequences with randomly shuffled word order (Gauthier & Levy, 2019; Chiang & Lee, 2020; Sinha et al., 2021). Lv et al. (2023) explore a fine-tuning objective with bidirectional attention and show that it can mitigate the reversal curse in the original synthetic setting from Berglund et al. (2023). However, they employ fixed masking rates. In addition to the standard objectives we explored, much recent work has gone into a variety of pre-training objectives including span-based and hybrid objectives (Joshi et al., 2020; Tay et al., 2022; Chowdhery et al., 2022). XLNet (Yang et al., 2020) utilizes a permutation language modeling objective, considering permutations of the input sequence during training. However, XLNet is not completely factorization-agnostic as it only predicts the last few tokens in each permutation.

Various benchmarks have been introduced to evaluate the reasoning capabilities of language models. Bachmann & Nagarajan (2024) present a study on the limitations of next-token prediction in capturing reasoning abilities, arguing that the standard autoregressive training objective hinders models' ability to plan. In a similar vein, Dziri et al. (2023) investigate the limits of transformer LLMs across three compositional tasks: multi-digit multiplication, logic grid puzzles, and a classic dynamic programming problem. Their findings suggest that transformer LLMs solve compositional tasks by reducing multi-step compositional reasoning into linearized subgraph matching, without necessarily developing systematic problem-solving skills. They also provide theoretical arguments on abstract multi-step reasoning problems, highlighting how autoregressive generations' performance can rapidly decay with increased task complexity.

## 6 Discussion and Future Work

**Limitations and Potential Extensions.** MLM-$\mathcal{U}$ has a much more challenging objective since we approximate all possible partitions of the input into context and predictions. Learning curves show delayed generalization, especially on backward samples. The main limitation of factorization-agnostic approaches is the optimization difficulty due to task complexity. Predicting one token ahead is far easier than predicting the last word of a novel with limited context, due to increasing entropy along longer horizons. This requires better schedules/curricula that smoothly interpolate the difficulty increase from next-token prediction to the highest-complexity factorization the model can handle.

This work highlights how alternative objectives can address some of the issues with current state-of-the-art language models, which rely on left-to-right autoregressive generative decoder pretraining. Despite impressive capabilities with increasing scales, there are concerns about reaching a plateau due to fundamental limitations, computational constraints, or data scarcity. We find that factorization-agnostic training can learn "more" from the same data in the context of reversal curse.

This presents a case for studying factorization-agnostic objectives and investing in approaches to scale them.

## Acknowledgements

We would like to thank Zeyuan Allen-Zhu, Sainbayar Sukhbaatar, Jason Weston, Olga Goloneva, and Léon Bottou for helpful discussions.

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

## A    Why does AR w/reverse sequences fail?

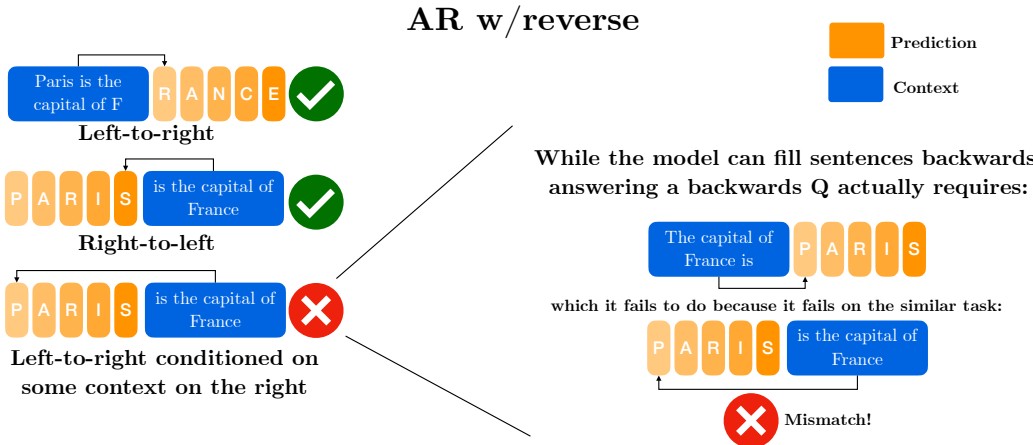

Figure 6: AR w/reverse cannot predict (left-to-right) entities that appeared on the left during training as it only learned to complete them from right to left. The two sequences in the bottom right indicate that backward retrieval is roughly equivalent to refactorizing the conditionals such that the entity of interest is predicted last conditioned on everything else. This is only approximate because answering a backward QA might require adding new tokens like *"The answer to the question is ..."* but we make a weak assumption that such differences are generally irrelevant compared to the entities and relations of interest.

## B    Permutation Language Modeling and Discrete State Diffusion

To illustrate the similarity between the diffusion loss and permutation language modeling, let's continue walking through our $D = 2$ example. Permutation modeling averages over factorizations $p(\boldsymbol{x}) = \frac{1}{2}p(x_2|x_1)p(x_1) + \frac{1}{2}p(x_1|x_2)p(x_2)$ and optimizes a lower bound on the likelihood

$$\log p(\boldsymbol{x}) \geq -\mathcal{L}_P = \frac{1}{2}(\log p(x_2|x_1) + \log p(x_1)) + \frac{1}{2}(\log p(x_1|x_2) + \log p(x_2)). \tag{6}$$

Finally, the diffusion model averages over masking rates $\frac{1}{2}(\log p(x_1) + \log p(x_2)) + \frac{1}{2}(\log p(x_1|x_2) + \log p(x_2|x_1))$ and optimizes

$$\log p(\boldsymbol{x}) \geq -\mathcal{L}_{\text{MLM}\mathcal{U}} = \frac{1}{2}(\log p(x_1) + \log p(x_2)) + \frac{1}{2}(\log p(x_1|x_2) + \log p(x_2|x_1)). \tag{7}$$

This is the same as Equation (6). This implies that the permutation language modeling and the absorbing state diffusion objectives are in fact the same. Though practically speaking, they may have very different implications.

## C    Summary of Tables

Table 4 shows a qualitative comparison of the optimization objectives explored on the different datasets in this paper. We conclude that MLM with a fixed masking rate mitigates the reversal curse due to its bi-directionality, but lacks generative quality and thus generally fails when having to provide longer answers. Also unsurprisingly, the left to right AR objective works well in the forward retrieval direction but is unable to answer backwards questions and has a hard time reasoning multiple tokens ahead to solve a task like graph traversal without intermediate supervision. Reversing the tokens can aid backwards retrieval for single token lookups, but fail otherwise. Reversing entities intuitively should be able to solve every retrieval task, but finding the right token permutation is a

Table 4: Summary of qualitative results, formatted as (forward)/(backward). Stargraph only has one direction.

| Task | MLM | MLM-$\mathcal{U}$ | AR | AR rev. | AR rev. ent. |
|------|-----|------|-----|---------|--------------|
| Retrieval | ✓/✓ | ✓/✓ | ✓/✗ | ✓/✓ | ✓/✓ |
| Relationship | ✓/✓ | ✓/✓ | ✓/✗ | ✓/✓ | ✓/✓ |
| BioS | ✗/✗ | ✓/✓ | ✓/✗ | ✓/✗ | ✓/✓ (Golovneva et al., 2024) |
| Wiki | ✗/✗ | ∼/∼ | ✓/✗ | ✓/✗ | – |
| Stargraph | ✓ | ✓ | ✗ | ✗ | ✓ (Bachmann & Nagarajan, 2024) |

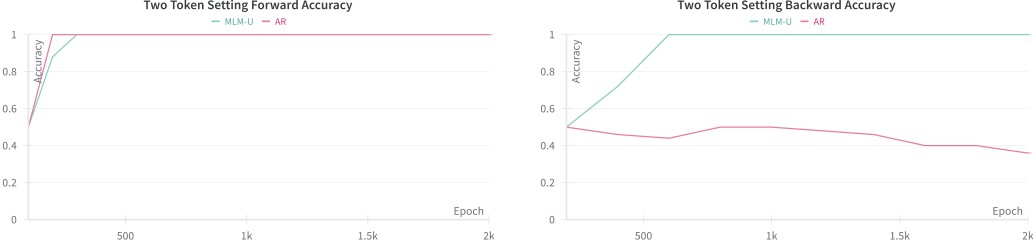

Figure 7: **Performance of MLM-U versus AR in the two-token setting**. We train both MLM-U and AR in a two-token variant of the retrieval task from from Section 3.1. We find MLM-U reaches 100% forward and backward whereas AR struggles to learn the backwards setting.

difficult task by itself. MLM-$\mathcal{U}$ averages over all possible prediction tasks that exist for a sequence given a tokenization and prevails in most our experiments. MLM-$\mathcal{U}$ displays the highest backwards retrieval capabilities in the most realistic Wikireversal benchmark, but the performance is not strong enough to qualitatively state success and we mark it with $\sim$ in Table 4. We hypothesize that the reason is increased task complexity requiring larger models. Notably, in Table 8 we show that MLM-$\mathcal{U}$ outperforms all other objectives when it has access to either the forward or backward type question. From there, it can generalize well to the other type.

We also experiment with a simple two-token setting comparing MLM-$\mathcal{U}$ and autoregressive next token prediction in Figure 7.

# D  Additional Tables

Table 5: Retrieval Task forward and backward per token accuracy of different training paradigms.

| | AR | AR w/reverse | MLM 15% | MLM 40% | MLM 85% | MLM-$\mathcal{U}$ | PLM |
|--|-----|--------------|---------|---------|---------|------|-----|
| Forward | **100** | **100** | 21 | 17 | 27 | **100** | **100** |
| Backward | 0 | 0 | 22 | 16 | 28 | **100** | **100** |

Table 6: BioS exact match accuracy for property retrieval in the backward direction (birth date to full name) and in the forward direction (full name to birthdate).

| | AR | AR w/reverse | MLM 15% | MLM 40% | MLM 85% | PLM | MLM-$\mathcal{U}$ |
|--|-----|--------------|---------|---------|---------|-----|------|
| Forward | **1.00** | **1.00** | 0.00 | 0.08 | 0.04 | **1.00** | **1.00** |
| Backward | 0.00 | 0.00 | 0.00 | 0.08 | 0.08 | **0.72** | 0.68 |

Table 7: Exact match QA accuracies for relationship tasks. Forward and backward accuracies are calculated normally, but due to the non-reciprocal relationship, a model that swaps the subject and object will make errors (e.g., inferring *B* is *A*'s child from *A* being *B*'s child). Entity reversal without a delimiter is marked with a*.

| | AR (entity) | w/reverse | AR (entity)* | w/reverse | MLM 15% | MLM 40% | MLM 85% | MLM-$\mathcal{U}$ | PLM |
|---|---|---|---|---|---|---|---|---|---|
| Forward | **100** | | 54 | | 24 | 77 | 2 | **100** | **100** |
| Backward | **100** | | 53 | | 19 | 35 | 1 | **100** | **100** |
| Incorrect Inference | 0 | | **44** | | 0 | 1 | 0 | 0 | 0 |

Table 8: Wikireversal task exact match QA accuracies. MLM-$\mathcal{U}$, MLM and AR are all 100M parameter models trained from scratch. (Right) uses different seeds for train test splits in forward and backward questions while (Left) uses the same seed. For MLM, we tried 15%, 40% and 85% masking rates and we present only the best models (15%). Details on hyperparameter selection can be found in Appendix E.3

| | Mistral 7B | MLM | MLM-$\mathcal{U}$ | AR | Mistral 7B | MLM | MLM-$\mathcal{U}$ | AR |
|---|---|---|---|---|---|---|---|---|
| Forward | **21** | 3.4 | 11 | 14 | 20 | 29 | **66** | 28 |
| Backward | 5.2 | 2.7 | **7.9** | 4.3 | 9.0 | 10 | **46** | 6.2 |

---

**Algorithm 1** Dataset Creation

---

**Input:** GenWiki Corpus $\mathcal{G} = \{(P, E, T)\}$
**Output:** QA Dataset $\mathcal{D} = \{(q, a, P)\}$
$\mathcal{D} \leftarrow \emptyset$
**for** $(P, E, T) \in \mathcal{G}$ **do**                   ▷ Each GenWiki sample
  **for** $(e_i, r, e_j) \in T$ **do**               ▷ Each relation triple
    **if** $e_i$ appears before $e_j$ in $P$ **then**       ▷ Forward relation
      $q_f \leftarrow F_r(e_i)$              ▷ Forward question
      $q_b \leftarrow B_r(e_j)$              ▷ Backward question
      $\mathcal{D} \leftarrow \mathcal{D} \cup \{(q_f, e_j, P), (q_b, e_i, P)\}$
    **end if**
  **end for**
**end for**
Filter $\mathcal{D}$ to keep unambiguous QA pairs       ▷ See filtering in Appendix E.1
Filter $\mathcal{D}$ to remove rare QA pairs where relation $r$ appears $< 500$ times

---

# E WikiReversal

We outline the setup for the WikiReversal dataset. After installing the GenWiki package https://github.com/zhijing-jin/genwiki, users can download and process GenWiki to form WikiReversal using:

```
gen_wiki_data = GenWikiReader()
gen_wiki_data.print_sample()
all_data = gen_wiki_data.read()
sample = all_data[3]

def check_valid(question):
    counts = 0
    for char in question:
        if char == "{":
```

```
        counts += 1
    if char == "}":
        counts -= 1
    if counts < 0:
        return False
return counts == 0

def check_valid_all():
    for key in questions_dict:
        if not check_valid(questions_dict[key]["forward"]):
            print(key, "forward")
        if not check_valid(questions_dict[key]["backward"]):
            print(key, "backward")

def is_forward(sample, graph_idx) -> int:
    e1 = sample['graph'][graph_idx][0]
    e2 = sample['graph'][graph_idx][2]
    i1 = text.find(e1)
    i2 = text.find(e2)
    if i1 == -1 or i2 == -1:
        return -1
    return int(i1 < i2)
```

### E.1    Filtering Ambiguous Samples

To mitigate ambiguity in the generated QA pairs, we filter the dataset to retain only $(e_i, r, e_j)$ triples where the $(e_i, r)$ and $(r, e_j)$ pairs are unique across the entire dataset. This ensures that each question has a single unambiguous answer. Algorithm 1 summarizes the dataset creation process.

### E.2    Examples from the Wikireversal dataset

Table 10 shows the relations present in the Wikireversal dataset. Table 9 shows examples of passages and corresponding forward and backward questions that are trained on. WikiReversal is filtered from GenWiki (Jin et al., 2020), a dataset based on Wikipedia released under a Creative Commons Attribution 4.0 International License.

### E.3    Details on Wikireversal training

To measure performance on the Wikireversal dataset, we split the available data into training and validation, where we include all passages and 80% of both forward and backward questions in the training set and 20% of questions in the validation set. We run a hyperparameter grid search over every objective. We sweep over feasible learning rates for all models and weight decay for all except for MLM-$\mathcal{U}$, where we haven't found weight decay to be effective so it is set to 0. We run sweeps for both different (Table 8 right) and same (Table 8 left) train test splits in forward and backward questions. GPT and BERT models have 12 layers with 12 heads and 768 embedding dimension for a total of 108 M parameters. The encoder-decoder model has 12 layers with 9 heads and 576 embedding dimension for a total of 109 M parameters. All models except Mistral were trained for 1500 epochs, and Mistral was trained for 200 to alleviate overfitting. The learning rates were warmed up for 1% of the training time in all cases. For Mistral, the LoRA $\alpha$ and $r$ parameters are set to 256 to sum to about 109 M trainable parameters. Learning rates are 5e-5 for MLM-$\mathcal{U}$ in both train split modes, 3e-4 for both BERT (MLM-15%) and GPT (AR) for both modes and 1e-4 for Mistral (AR). The best weight decays are 1e-2 for both BERT and GPT, and no weight decay for LoRA on Mistral. No dropout was used. All models were trained with AdamW with default $\beta$ parameters.

Table 8 shows that MLM-$\mathcal{U}$ outperforms other objectives, achieving 66% and 46% accuracy on forward and backward questions, respectively. Using different seeds for train/test splits in forward and backward directions (right section) allows bidirectional models to learn to answer questions from both the passage and the question itself, explaining MLM-$\mathcal{U}$'s significant improvement. AR

Table 9: Examples from Wikireversal

| Passage | Forward Q | Backward Q |
|---|---|---|
| Agostino Magliani ( 23 July 1824 – 20 February 1891 ), Italian financier, was a native of Laurino, near Salerno. | Where was Agostino Magliani born? | Who was born in Laurino? |
| Zhou Yongkang has two sons, Zhou Bin and Zhou Han, with his first wife, Wang Shuhua, whom he met while working in the oilfields of Liaoning province. | Who is Zhou Yongkang's spouse? | Who is married to Wang Shuhua? |
| The total area of Mitan-myeon is 109.74 square kilometers, and, as of 2008, the population was 1,881 people. | What is the total area of Mitan-myeon? | Which populated place has a total area of 109.74? |
| Mohammad Ali Araki was born on 1894 in Arak, Iran. He started his education from Arak Hawza. Grand Ayatollah Haeri allowed him to wear the turban and robe because qualified individuals were limited. Also, Araki studied many years in Yazd Hawza. | What title does Mohammad Ali Araki hold? | Who holds the title of Grand Ayatollah? |
| Tibor Navracsics (born Veszprém, Hungary, 13 June 1966) is a Hungarian lawyer and politician, who served as Minister of Foreign Affairs and Trade from June to September 2014. | What region is Tibor Navracsics located in? | What or who is located in the Veszprém region? |
| WWWX ( 96.9 FM, "96.9 The Fox" ) is an Alternative rock formatted radio station licensed to Oshkosh, Wisconsin, that serves the Appleton-Oshkosh area. | What is WWWX's alias? | Whose alias is 96.9 The Fox? |

performs poorly on backward questions due to the "reversal curse". MLM and Mistral 7B show intermediate performance. Although Mistral 7B uses $\sim$ 100M LoRA parameters, fewer than the other models, this setup mimics common fine-tuning recipes. Naturally, models trained from scratch do not learn general language modeling capabilities.

Table 10: Relations in Wikireversal

| Attribute | Count |
|---|---|
| birthPlace | 6100 |
| birthName | 5018 |
| alias | 3745 |
| location | 2532 |
| deathPlace | 2064 |
| title | 1923 |
| city | 1871 |
| populationTotal | 1651 |
| owner | 1328 |
| name | 1274 |
| spouse | 1163 |
| isPartOf | 1000 |
| type | 920 |
| office | 893 |
| associatedBand | 762 |
| associatedMusicalArtist | 756 |
| synonym | 743 |
| knownFor | 729 |
| artist | 724 |
| PopulatedPlace/areaTotal | 719 |
| birthDate | 672 |
| ground | 670 |
| occupation | 665 |
| place | 631 |
| address | 631 |
| family | 589 |
| hometown | 559 |
| region | 551 |
| developer | 541 |
| label | 538 |
| writer | 517 |
| total count | 42479 |

# F    Delayed Generalization in Language Modeling

We include accuracy curves for training with MLM-$\mathcal{U}$ for both Bios and WikiReversal in Figure 8. We see the model is able to gradually learn both the forward and backward questions throughout training. For Bios, unlike the forward questions which saturate much more quickly, the backward accuracy still shows an upward trend after training for 20k optimization steps. We observe a similar trend in the delayed generalization in WikiReversal for both forward and backwards questions even after training for 300k optimization steps. These results empirically demonstrate that the MLM-$\mathcal{U}$ objective, which requires modelling all possible factorizations of an input into context and predictions, is a more challenging task that exhibit delayed generalization relative to standard next-to-prediction training.

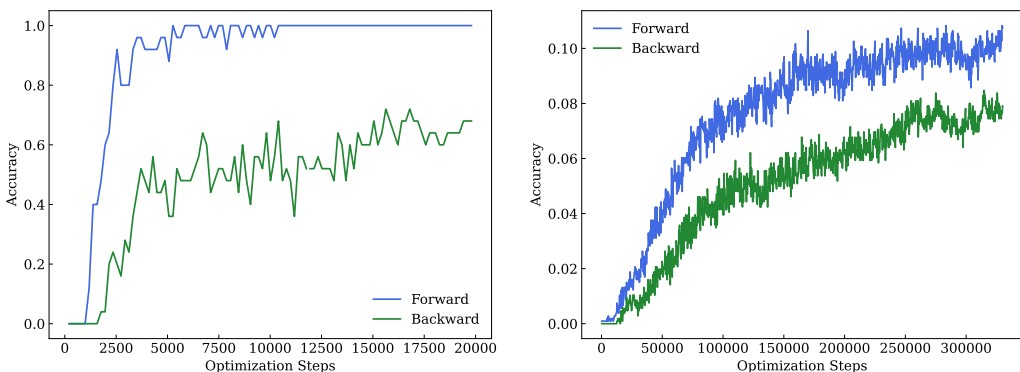

Figure 8: Accuracy in Forward/Backward Questions on the Bios dataset (left) and the Wikireversal dataset (right)

# G    Architecture Details

The Encoder-Decoder architecture used to train the MLM-$\mathcal{U}$ objective is modeled with ideas from XLNet Yang et al. (2020) in mind in order to support different attention/masking strategies including permutation language modeling. The encoder has GPT-like blocks and works with RoPE as positional bias. The decoder also has GPT-like blocks, but it cross-attends over keys and values from the corresponding encoder layer, also via a RoPE bias. The decoder input contains the same learnable embedding for all tokens, such that only the positional bias defines the initial attention pattern. This idea comes from XLNet's positional attention stream. In left to right AR training mode, both encoder and decoder use a causal attention mask. In MLM-X modes, a fraction of inputs are masked before given to the model and neither decoder nor encoder attend over the masked tokens. All inference is performed in left-to-right AR fashion.

# H    Compute Requirements

Models were trained on 64 NVidia V100 and A100 GPUs with supporting Intel(R) Xeon(R) Gold 6230 CPUs. From conception to finalization of this paper we trained about 2000 models. The computationally most expensive runs were on the BioS and the Wikireversal dataset. Those comprised about 300 runs with on 8 GPUs for around a day per model. About 30 Mistral models were trained on 32 GPUs for about a day per model. We also compare the convergence relative to GPU hours for MLM-$\mathcal{U}$ to AR in Figure 9.

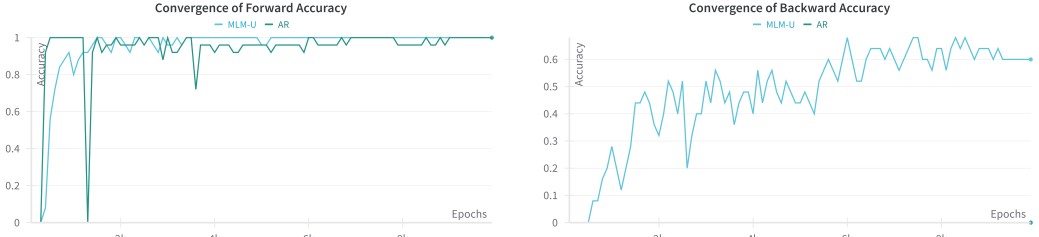

Figure 9: **Comparing the of convergence for MLM-U versus AR on retrieval**. We found MLM-U took 558.80 minutes versus 559.45 for AR to train on 8 V100 GPUs. Although we observe faster forward saturation for AR, convergence is much noisier and the AR model is not able to learn the backwards task, whereas MLM-U is.

