# OpenReview forum: "The Factorization Curse: Which Tokens You Predict Underlie the Reversal Curse and More"
_NeurIPS.cc/2024/Conference — NeurIPS 2024 poster_

### Official Review · Reviewer_kVoQ · 2024-07-09

**Soundness:** 3
**Presentation:** 4
**Contribution:** 3
**Rating:** 7
**Confidence:** 3

**Summary:**

This paper focus on the "reversal curse", where models struggle to recall information when probed in a different order than encountered during training. The authors propose reframing this issue as the "factorization curse", which is a failure of models to learn the same joint distribution under different factorizations. They support it with experimental evidence, providing also a new dataset WikiReversal, to evaluate this problem and explore solutions. Authors suggest to use factorization-agnostic training objectives, which can significantly mitigate these issues.

**Strengths:**

1. Authors introduce novel concept of  "factorization curse", broadening the understanding of why models fail in information retrieval.
2. The development of WikiReversal, based on Wikipedia knowledge graphs, is a valuable contribution, offering a more realistic benchmark for evaluating model performance in information retrieval tasks.
3. Authors provide extensive experiments, proving the fact that "factorization curse" exists both in toy examples and real cases.
4. They suggest a factorization-agnostic training objective that showed good performance in mitigating such issues, while preserving satisfactory quality in forward queries.

**Weaknesses:**

1. While the paper presents a variety of experiments with different training objectives, they experiment with small model sizes (100M parameters). It is not clear how well this approach could be scaled on bigger models.
2. Authors didn't clearly analyze whether proposed training objective will hurt the quality on the standard benchmarks. For example, it would be interesting to see if the quality of the open generation drops with new training objective.

**Questions:**

1. Do the quality of generation drops if we use proposed method to train the model?
2. Will you give the open access to the provided WikiReversal?
3. What are the computational costs associated with factorization-agnostic training, and how do they compare to traditional training objectives?

Also some typos:

Line 83 "are: missed in “Note that there many”

**Limitations:**

1. The experiments are primarily based on the WikiReversal testbed, which, while realistic, is still limited to the structure and content of Wikipedia knowledge graphs.
2. Limited scales of models, the training was done only on small models, which could be justified for retrieval itself, but need to be addressed for scalability of approach

---

> ### Author Rebuttal · Authors · 2024-08-06
>
> Thank you for providing valuable feedback to improve our work. We appreciate you finding the factorization curse a novel concept to explain model information retrieval failures. We are happy you appreciate the development of WikiReversal and our extensive experiments. Based on your suggestions we ran an additional experiment to measure the computation costs and convergence tradeoffs of MLM-U. We outline these new experiments and answer your remaining questions below.
>
> ### Scale and performance on standard benchmarks
> Given that our objective is not to train a general purpose language model, we do not benchmark standard language benchmarks as here we isolate only information retrieval. This helps us control the experiments. Albeit limited by relatively small context windows in our knowledge retrieval tasks, we have confirmed that generated text is grammatically coherent and looks like proper English.
> Overall in our experiments, we focus on whether alternative learning objectives, namely factorization-order agnostic objectives can overcome information retrieval failures such as the reversal curse.
> For all information retrieval benchmarks considered, we found a modestly sized model was able to perform quite well. Even for the more challenging WikiReversal a much smaller model outperformed Mistral-7B finetuned on the same data—demonstrating the advantage of factorization-agnostic learning objectives such as MLM-U. Of course, we agree scaling such models further would be interesting and we are exploring this direction for future work.
>
> ### Access to WikiReversal
> We agree it is important for the research community to build on the WikiReversal benchmark. Fortunately, the underlying data is already openly accessible via GenWiki https://github.com/zhijing-jin/genwiki with an openly accessible (Creative Commons) license. While WikiReversal should be reproducible from the details outlined in Section E (specifically Alg. 1), we do plan to include the exact scripts we use for researchers to download and parse the dataset to form WikiReversal in order for the community to study the reversal curse on more realistic natural text.
>
> ### Computational Costs and Convergence
> Thank you for raising this point. We agree comparing the computational costs for each training approach is important. Based on this suggestion, we analyzed the runtime and convergence rates for both MLM-U and standard autoregressive (AR) training.  We benchmarked two parameter matched models on the retrieval task dataset from Section 3.1. We found MLM-U and AR exhibited comparable computational costs per training step:  558.80 minutes versus 559.45 (AR) on 8 V100 GPUs after 1k epochs of training.
>
> To better understand the convergence tradeoffs we also include a comparison of the forward and backward accuracy curves for MLM-U versus AR in Figure 2 of the supplementary rebuttal PDF.  As expected we found the forward loss for the AR converged more quickly with forward saturating after 1600 epochs versus MLM-U, which saturated after 3999 epochs of training, although we observed a much smoother convergence for MLM-U even for forward. Of course, the AR model despite the faster forward convergence was not able to improve the backward accuracy. Based on your suggestion we plan to include a more thorough discussion of the computational cost and convergence in the revised draft. We’ve included both forward and backward converge plots in Figure 3 the rebuttal PDF for your reference as well. Note also appendix F, which talks about the delayed generalization speed for the backward direction.
>
> We’d like to thank you for the thoughtful questions and suggestions. We hope the additional experiments and clarifications have addressed your questions. We remain available for any further discussion or questions.

---

> > ### Comment · Reviewer_kVoQ · 2024-08-12
> >
> > Thank you for your quick and very detailed reply. I believe the score is already high enough, but you have fully answered all the questions.
> > Best wishes to your paper, and let me know if I can be of any help.

---

> > > ### Author Response · Authors · 2024-08-14
> > >
> > > Thank you very much!

---

### Official Review · Reviewer_bZEp · 2024-07-10

**Soundness:** 3
**Presentation:** 4
**Contribution:** 3
**Rating:** 7
**Confidence:** 3

**Summary:**

This paper addresses the reversal curse, where models trained on a relation in one direction (e.g. "A is the capital of B") cannot answer questions about the relation worded in the reverse order (e.g. "B's capital is [?]"). The authors frame this as a subcase of a broader problem, where models trained with a causal learning objective do not learn to assign equal probability to different factorizations of the same sequence, which the paper terms the factorization curse. To address the factorization curse, the authors propose using MLM-U, a modified MLM that varies both the location and size of the masked segments. The authors show that models trained with this modified objective do not exhibit the same reversal curse and analyze the behavior of models trained with causal, MLM, and MLM-U objectives.

**Strengths:**

S1. The paper tackles a well-defined problem from an interesting angle, providing both theoretical and empirical evidence for its claims. The paper is well-written.

S2. The connection between PLM and discrete state diffusion here is an interesting insight and well-explained.

S3. WikiReversal is an interesting idea for a dataset and looks like it will be a great resource for the community, especially for work aiming to study this reversal phenomenon. It is documented well.

**Weaknesses:**

W1.  In a two-token setting, you can (at a high but not intractable computational cost) compute the right-to-left factorization of a causal model through enumeration. It would be interesting to compute the left-to-right and right-to-left factorizations for some two-token sequences at test time for the models presented here. Do the causal models clearly suffer from a discrepancy of joint probabilities in the two-token case? Does the MLM-U training mitigate the two-token factorization curse? It seems likely the answer is yes, but it would strengthen the paper to confirm this empirically. (And in a 2-token setting, it seems that MLM would also mitigate the factorization curse-- does this happen in practice?)

W2. Comparing BERT-like MLM to encoder-decoder MLM-U or GPT-like AR seems to be a bit of a strange comparison. It would seem more natural to use a BART-like encoder-decoder for the MLM example (though I recognize that the causal attention in the decoder may be slightly confounding, so I think both settings have merit; still, it would be good to *see* both to be sure).

W3. The architecture details for MLM-U, while not the key argument of the paper, are only sparsely explored. It is not clear how much of the benefit comes from using the new architecture design versus using the MLM-U objective. And since Appendix G suggests that this architecture can handle both MLM-X and AR training, why not just train all models using this architecture and vary only masking strategy?

**Questions:**

Q1. In appendix B, you state that the PLM and absorbing state diffusion objectives are “the same” but “practically speaking […] may have very different implications”. Can you elaborate on this point?

Q2. Why use BERT rather than BART (or another encoder-decoder) as the choice for MLM?

Q3. You argue that at least some of the benefits of MLM-U over MLM arise from the ability to fully mask out multi-token entities. Could you demonstrate that, given a task where the entities are all single-token, the benefit of MLM-U is marginal or non-existent? I’m thinking of a version of the retrieval task in 3.1 where the key-value sequences are all of length 1.

typos/small notes:
* line 79: GPT-style citation should probably be GPT-2 or GPT, not GPT-4.
* line 83: “Note that there *are* many factorizations”
* Table 1: very small point, but I would have expected the entry with the asterisk to be the entry *with* delimiter tokens.
* I think the factorization curse section could emphasize even more that the ordering of the tokens in the input is not altered when calculating the different factorizations. This is stated in the text (lines 83-85 and 92), but given that the section begins by discussing causal models (and, like a lot of us, I work almost exclusively with causal models these days!), it still required a second read to parse exactly what was going on.

**Limitations:**

Limitations seem appropriate.

---

> ### Author Rebuttal · Authors · 2024-08-06
>
> We thank the reviewer for carefully considering our work. We are glad you found the paper well written and that we tackle a well-defined problem with both theoretical and empirical insights. We are especially happy you appreciated the effort we put into crafting a more realistic benchmark for the reversal curse with WikiReversal. We appreciate the suggestions you provide to improve the work and have performed additional experiments based on these suggestions, which we address below.
>
> ### Two-Token (W1)
>
> Thank you for the suggestion to experiment with a two-token setting to assess whether MLM-U training mitigates the two-token factorization curse. We ran a new experiment comparing standard left-to-right AR training (labeled GPT in the legends) and MLM-U. Specifically, we train both MLM-U and AR in a two-token variant of the retrieval task from Section 3.1 trained for 2k epochs. We find MLM-U reaches 100% forward and backward whereas AR struggles to learn the backwards setting, reaching only 12% accuracy after 2k epochs of training. We’ve attached plots demonstrating the forward and backward accuracy of each model throughout training in Figure 1 of the supplementary rebuttal PDF. Please let us know if this answers the question you had in mind.
>
> ### Encoder-decoder architecture comparisons (W1, W2, Q2)
>
> While we agree that training all models with encoder-decoder and then varying only the masking strategy is one valid strategy, we wanted to give each masking strategy its “best shot”. We argue that it is most fair to have each masking strategy compete with a corresponding architecture that is known to work well with it. Otherwise one could, for instance, claim that the encoder-decoder is not well suited (or not properly optimized) for AR.
> (FYI, earlier on in our research investigations we experimented with both encoder-decoder and BERT-style encoder only architectures for MLM. There we found BERT-style encoder architectures to perform better with MLM-X objectives.)
>
>
> ### MLM versus MLM-U (Q3)
>
> We would like to clarify that both MLM and MLM-U are capable of masking and predicting multi-token entities (unless of course the entity is longer than the masking ratio). The main advantage of MLM-U as shown in Figure 2 stems from its ability to handle context-dependence of variable lengths.
>
> ### Practical differences between PLM and MLM-U (Q1)
> While the objectives are theoretically equivalent, the implementation of XLNet is not fully factorization agnostic in practice. As described in Section 2.3 of XLNet  https://arxiv.org/abs/1906.08237 for "practical reasons they end up training with a permutation on the last few tokens only." This results in a model that is not fully factorization-agnostic.
> Additionally, in practice, we do not average over all permutations or all masking rates, but only a randomly chosen subset. This might induce more practical differences.
>
> Thank you for the proposed clarifications and language suggestions. We’re very glad you point out “the input is not altered when calculating the different factorizations.” We agree we’ll emphasize this key point much more prominently in the writing. We remain available for further questions and thank you again for all the effort you put into this review.

---

> > ### Comment · Reviewer_bZEp · 2024-08-12
> >
> > Thanks for the detailed reply and additional experiment! The two-token setting you ran is quite interesting, though not what I originally had in mind-- I was thinking of a demonstration of the issue for causal models. In lines 86-95, you describe the two-token case in detail; for a causal model, you would generally only be able to compute $p(x_2|x_1)p(x_1)$. However, in a 2-token case, it's possible to fix a value for $x_2$ and enumerate all possible pairs $x_1 x_2, x_1 \in V$, so that you can compute $p(x_1|x_2)$, allowing you to directly compute the "backwards" factorization $p(x_1|x_2)p(x_2)$. I think it would be a nice empirical demonstration to compute "forwards" and "backwards" factorization of a few sequences under the causal, MLM, and MLM-U models, to show that the causal model has a much higher difference in probability between these two factorizations.
> >
> > However, this is not necessary to the paper-- more of a demonstration-- and I appreciate the effort that went into the rebuttal as a whole, so I will raise my score 6 -> 7. I think this is a good paper!

---

> > > ### Author Response · Authors · 2024-08-14
> > >
> > > Oh, sorry for the confusion about the experiment!
> > > In any case, thank you very much for your evaluation!

---

> > > ### Author Response · Authors · 2024-08-14
> > >
> > > if we computed p(x1|x2) through p(x1,x2) and the marginals, wouldn't we circumvent the issue? I don't think we have time to discuss anymore, but we shall think about this experiment for a possible camera ready version. Thank you for the suggestion!

---

### Official Review · Reviewer_Z95k · 2024-07-14

**Soundness:** 3
**Presentation:** 3
**Contribution:** 3
**Rating:** 6
**Confidence:** 3

**Summary:**

The paper extends the idea of the "Reversal Curse" from prior work and proposes ways to mitigate it by finetuning LLMs with a different objectives. To recall, the reversal curse is formulated roughly as follows: a model, when **finetuned,**(i.e. not prompted) in A is B statement, does not automatically generalize to B is A. Authors generalize this, from a probabilistical point of view, into the inability to generalize between different factorization orders on the joint text distribution. Authors then hypothesize that this can be alleviated by what they call "permutation-agnostic training" - training techniques popular among LLM that require predicting tokens in a varying order. Authors consider a number of such techniques based on prior work on encoder pretraining and find that MLM-u offers a consistent solution to the reversal/factorization curse, while many other intuitive solutions don't. Authors experiment using two relatively smaller language models (GPT-2 and Mistral 7B) on tasks like simple retrieval, understanding non-reciprocal relationships  (between sets/statements), WikiReverasal. The paper also analyzes the representations learned after the model is fine-tuned with the proposed approach.

**Strengths:**

1. The paper proposes (seemingly) a solution that covers one of the reasons why LLMs hallucinate. Since LLM hallucination is one of the main roadblocks to their greater adoption, this is an impactful problem to address.

2. The experiments appear sound: authors evaluate on a diverse set of tasks, using two different LLMs. This eliminates the possibility of a false positive. However, there is still a direction of scaling to larger models (e.g. 70B and above) and checking against possible changes in the efficacy of the proposed solution.

3. The proposed method relies on fine-tuning, not training a model from scratch, and thus can be broadly applied to existing models

4. The paper is generally well written, has a clearly stated hypothesis and is overall easy to follow, if a bit unconventionally structured. Minor typos and clarification requests (below) do not affect the overall presentation.

Also, a minor but pleasant advantage of this paper is that they openly declare that their solution is a clever reuse of an existing method. Many works I read in the past instead  choose to slightly modify the approach and declair it a newly proposed method. I count your choice as a minor advantage because it reduces the research debt, i.e. how many methods does a newcomer need to learn to meaningfully contribute to this area.

**Weaknesses:**

### Side-effects can be explored better

My main concern with the paper is that , while there is great attention to how fine-tuning combats reversal curse, but the side-effects of such fine-tuning are arguably not explored enough.


**In other words, does fine-tuning Mistral affect its accuracy on other, unrelated tasks? If yes, what trade-off points are there?**

To test that, once can use LM Eval Harness for a selection of such tasks ( https://github.com/EleutherAI/lm-evaluation-harness ). If not familiar with the tasks, please use the ones commonly reported upon popular LLM releases (e.g. see Table 20 in the Llama 2 paper https://arxiv.org/pdf/2307.09288 ) or choose your own tasks. Another direction would be to evaluate the overall quality of your model's generated text, whether with human evaluation (local/MTurk/Toloka/...) or, as an Ersatz, with another LLM (as in https://arxiv.org/pdf/2304.03277 )

This can make a difference between a "free" solution to the curse and the one that comes at too great a cost got most practitioners -- or something between the two. As such, the paper would greatly benefit from understanding these trade-offs, or knowing that they don't exist.


### XLNet the Baseline

You (rightfully) spend a lot of Section 2.2 on describing XLNET (alongside MLM-U), but then never compare against that as a baseline.

While you offer some algorithmic reasons MLM-U could be better, but to dismiss a relevant algorithm as a baseline, one usually needs more evidence, e.g. showing that it is infeasible to use or proving that it is guaranteed to be worse. I could not find such an argument in the paper. If you have it, please direct me to it. If not, the paper could be improved by actually comparing to XLNET, at least as a side-experiment on a subset of tasks, so that the reader better understands your choice of MLM-U.

**Questions:**

### Does this scale?

In your work, you test the proposed solution on GPT-2 and Mistral 7B. While the latter is undoubtedly an LLM, it is still curious if your approach generalizes to more capable LLMs. Note that I am not asking you to run all the experiments with a larger model, but even a selection of such experiments would improve the paper, particularly for practitioners.

If your main concern is GPU size, it should be possible to fine-tune relatively larger models using QLoRA ( https://github.com/artidoro/qlora ) or ReLoRA ( https://arxiv.org/abs/2307.05695 ). The former can fine-tune a 65B model on a 48GB GPU or multiple consumer gpus with sharding. The latter amounts of running the formal, then merging the adapters into LLM parameters and running again, for several such loops. Note that there may be a potential confounder of LoRA adapters vs regular fine-tuning. If you care to disentangle these effects, one possible way is to first check if LoRA finetuning can lift the curse for Mistral 7B, and if it does, try for larger ones and compare against that.




> (L43) to learn learn logical implications,

Possibly an extra “learn” (typo)


> (L174-175)

there is a paragraph missing line numbers, likely due to excessive negative vertical space. In that paragraph, and in the unnamed equation below, you refer to tokens as $t_1, t_2$, etc. In turn, your text up to this point refers to tokens as $x_0, x_1$, etc Moreover, $t$ is explicitly reserved for token index (L80). Unless there is a specific reason for this, the presentation would be improved if you untangle the notation around $x$ and $t$.

**Limitations:**

To the best of my knowledge, authors have sufficiently addressed limitations of their work

---

> ### Author Rebuttal · Authors · 2024-08-06
>
> We are glad you regard reliable knowledge retrieval as an impactful problem and view MLM-U as a consistent solution to the factorization curse. We are happy that you found the paper to be well-written and suggested several useful pieces of feedback. We address each below:
>
> ### Finetuning
> First, we’d like to clarify MLM-U models in our experiments are trained from scratch—not finetuned. We found training from scratch directly on the downstream dataset (Tokens, BIOS, Wikigraph in Section 3.1 and 3.2) was sufficient for competitive performance or to solve the benchmark entirely without the need for pretraining. Given we’re training from scratch, we do not measure performance on side-effects related to standard benchmarks (though we do carefully check forward standard accuracy as well). However, we absolutely agree exploring MLM-U for finetuning would be a valuable and interesting direction we hope to tackle in an upcoming publication. For finetuning, side-effects in terms of standard benchmarks would then become paramount to understand the tradeoff between finetuning reliability.
>
> ### XLNet Baseline
> We absolutely agree XLNet is an important baseline to consider among factorization agnostic approaches. We present XLNet in Section 2.1 under the heading Permutation Language Modelling (PLM) where we describe the method as a baseline. In addition, we also draw a connection between permutation language modeling (such as XLNet) and discrete state diffusion in Section 2.2. We agree the heading and presentation can be made more clear to indicate we are in fact referring to XLNet in this section. The primary reason we chose to focus on MLM-U in the main tables is that the implementation of XLNet is not fully factorization agnostic: as described in Section 2.3 of XLNet  https://arxiv.org/abs/1906.08237 for "practical reasons they end up training with a permutation on the last few tokens only." Nevertheless we do provide additional comparisons to XLNet in Appendix Tables 5, 6, and 7 studying BIOS, QA relations, and synthetic tokens.  Based on your suggestions, we’ll rework the writing to ensure XLNet results are presented more clearly.
>
> ### Scaling MLM-U
> For fair comparisons to prior work in order to ensure our gains were the result of the proposed learning objective (and not mere differences in model scale), we chose to perform experiments using model scales from prior work ( retrieval task from Goloveneva et al. 2024 and BioS from Allen-Zhu et al. 2023).  We compare the effect of adjusting the learning objective only, keeping the model architecture fixed, to prior explorations that modify the training data (see Table 2 AR w/reverse for example).
> To push the experimental setting further towards larger realistic graphs with known entities we also develop a benchmark based on natural text from Wikipedia and naturally occurring entity relations. In this setting, we found even relatively small models trained with MLM-U performed remarkably well at resolving the reversal curse gap between forward and backward accuracy. Specifically, we found training from scratch only on the WikiGraph data with the MLM-U objective outperformed Mistral-7B, a model that’s 70x larger, after finetuning on the same data.
>
>
> Finally, we very much appreciate your attention to the typos, spacing, and notation suggestions. We’ve addressed each to sharpen the presentation of the work.

---

> > ### Comment · Reviewer_Z95k · 2024-08-14
> > **On Author Response**
> >
> > I thank the authors for answering my questions and clarifying some of my concerns. I still recommend that the paper should be accepted, and I am increasing my score by a notch.

---

> > > ### Author Response · Authors · 2024-08-14
> > >
> > > Thank you very much for your evaluation!

---

### Author Rebuttal · Authors · 2024-08-07

We’d like to thank reviewers for their high-quality feedback and thoughtful suggestions for our work. We very much appreciate reviewers noted the importance of reliable knowledge retrieval in language models noting “LLM hallucination is one of the main roadblocks to their greater adoption, this is an impactful problem to address”—Z95k. We’re glad reviewers appreciated our insight into the role of factorization in the reversal curse, noting we “introduce novel concept of ‘factorization curse’, broadening the understanding of why models fail in information retrieval”—kVoQ. We’re also glad reviewers found our experimental setup “sound” (Z95k) “realistic” and “extensive” (kVoQ) with several reviewers noting the value of WikiReversal to the research community. Finally, we’re glad several reviewers found our paper to be “well-written” (Z95k, bZEp) and our solution, MLM-U, to offer “a consistent solution to the reversal/factorization curse, while many other intuitive solutions don't”—Z95k.

Reviewers provided useful feedback on the scalability of the method, standard benchmark performance, computational tradeoffs, and suggestions to better isolate the effect of architecture as well as a new interesting two-token setting. We’ve made a considerable effort to incorporate this feedback with clarifications and three new experiments based on reviewers’ suggestions for which we’ve attached results in the supplementary rebuttal PDF. In summary, we have

- **Clarified scalability comparisons and benchmarks**: We ensure model scales were comparable to those used in prior work for tasks in section 3.1 such as BIOS. We also clarified for the WikiReversal experiments the MLM-U objective trained from scratch outperformed Mistral-7B, a model that’s 70x larger, after finetuning on the same data. Given our objective is not to train a general purpose language model, we do not benchmark standard language benchmarks as here we isolate only information retrieval when models are trained from scratch using MLM-U.

- **Compared in new experiments MLM-U versus AR training in the two-token setting proposed by reviewer bZEp (Figure 1 in PDF)**: We find MLM-U reaches 100% forward and backward accuracy whereas AR struggles to learn the backwards association in the two-token setting.

- **Measured the Computational Costs and Convergence of MLM-U (Figure 2 in PDF)**: We analyzed the runtime and convergence rates for both MLM-U and standard autoregressive (AR) training.  We benchmarked two parameter matched models on the retrieval task dataset from Section 3.1. We found MLM-U and AR exhibited comparable computational costs. While AR forward accuracy converges faster, MLM-U exhibits smoother convergence for the forward accuracy and is able to learn both backward and forward associations, whereas AR struggles to learn the backwards association.

After incorporating these new experimental results and suggestions, thanks to reviewers’ feedback, we believe the quality of our submission has improved. We hope the factorization curse illustrates the importance of factorization-agnostic learning objectives for reliable knowledge retrieval. We believe together with our realistic WikiReversal benchmark, the factorization curse and proposed solution would be a valuable contribution to the research community for advancing the reliability of knowledge retrieval.

---

### Decision · Program_Chairs · 2024-09-25

**Decision:**

Accept (poster)

**Comment:**

This paper examines the problem of reversal curse in LLMs by reframing it as a factorization curse —  a failure of models to learn the same joint distribution under different factorizations. Through a series of controlled experiments with increasing levels of realism, the paper finds that reliable information retrieval is an inherent failure of the next-token prediction loss widely used to train LLMs. It further shows that the problem cannot be easily addressed some several simple fix approaches. It further reveals a potentially promising direction to address this limitations by using factorization-agnostic objectives.

All the reviewers appreciate the solid contribution of this paper, such as a novel formulation of the reversal problem as a factorization curse. The paper is generally well written with solid experiments. The authors are encouraged to take the reviewers’ feedback into account when revising the paper.